# Decreased Haemoglobin Level Measured at Admission Predicts Long Term Mortality after the First Episode of Acute Pulmonary Embolism

**DOI:** 10.3390/jcm11237100

**Published:** 2022-11-30

**Authors:** Aleksandra Justyna, Olga Dzikowska-Diduch, Szymon Pacho, Michał Ciurzyński, Marta Skowrońska, Anna Wyzgał-Chojecka, Dorota Piotrowska-Kownacka, Katarzyna Pruszczyk, Szymon Pucyło, Aleksandra Sikora, Piotr Pruszczyk

**Affiliations:** 1Department of Internal Medicine & Cardiology, Medical University of Warsaw, 02-091 Warsaw, Poland; 21st Department of Radiology, Medical University of Warsaw, 02-091 Warsaw, Poland; 3Department of Hematology, Institute of Hematology and Transfusion Medicine, 02-776 Warsaw, Poland

**Keywords:** pulmonary embolism, follow-up after pulmonary embolism, long term mortality after pulmonary embolism, predictors of survival after pulmonary embolism

## Abstract

Background: Decreased hemoglobin concentration was reported to predict long term prognosis in patients various cardiovascular diseases including congestive heart failure and coronary artery disease. We hypothesized that hemoglobin levels may be useful for post discharge prognostication after the first episode of acute pulmonary embolism. Therefore, the aim of the current study was to evaluate a potential prognostic value of a decreased hemoglobin levels measured at admission due to the first episode of acute PE for post discharge all cause mortality during at least 2 years follow up. Methods: This was a prospective, single-center, follow-up, observational, cohort study of consecutive survivors of the first PE episode. Patients were managed according to ESC current guidelines. After the discharge, all PE survivors were followed for at least 24 months in our outpatient clinic. Results: During 2 years follow-up from the group of 402 consecutive PE survivors 29 (7.2%) patients died. Non-survivors were older than survivors 81 years (40–93) vs. 63 years (18–97) *p* < 0.001 presented higher sPESI 2 (0–4) vs. 1 (0–5), *p* < 0.001 driven by a higher frequency of neoplasms (37.9% vs. 16.6%, *p* < 0.001); and had lower hemoglobin (Hb) level at admission 11.7 g/dL (6–14.8) vs. 13.1 g/dL (3.1–19.3), *p* < 0.001. Multivariable analysis showed that only Hb and age significantly predicted all cause post-discharge mortality. ROC analysis for all cause mortality showed AUC for hemoglobin 0.688 (95% CI 0.782–0.594), *p* < 0.001; and for age 0.735 (95% CI 0.651–0.819) *p* < 0.001. A group of 59 subjects with hemoglobin < 10.5 g/dL showed mortality rate of 16.9% (OR for mortality 4.19 (95% CI 1.82–9.65), *p*-value < 0.00, while among 79 patients with Hb > 14.3 g/dL only one death was detected. Interestingly, patients in age > 64 years hemoglobin levels < 13.2 g/dL compared to patients in the same age but with >13.2 g/dL showed OR 3.6 with 95% CI 1.3–10.1 *p* = 0.012 for death after the discharge. Conclusions: Lower haemoglobin measured in the acute phase especially in patients in age above 64 years showed significant impact on the prognosis and clinical outcomes in PE survivors.

## 1. Introduction

Acute pulmonary embolism (PE) is not only life-threatening disease in the acute phase but also affects long term prognosis of PE survivors. PE patients may also experience serious adverse events at long term, in particular in the first years after PE diagnosis especially recurrent venous thromboembolism (VTE) and chronic thromboembolic pulmonary hypertension (CTEPH) [1]. Importantly PE survivors have an increased all-cause mortality risk [2]. Therefore, in order to optimize patients follow-up after acute PE it is recommended to ensure an integrated patient care after PE which should include interdisciplinary standardized management and treatment [1,3]. Clinical indices including pulmonary embolism severity index assessed in the acute phase were reported to predict accurately long-term mortality [4]. Notably, widely available biomarkers such as NT-proBNP and hemoglobin concentration were reported to predict long term prognosis not only in patients after congestive heart failure [5,6] or coronary artery disease [7], but also after exacerbation of chronic obstructive pulmonary disease or even pneumonia [8,9]. We hypothesized that hemoglobin levels may be useful for post discharge prognostication after the first episode of acute pulmonary embolism. Therefore, the aim of the current study was to evaluate a potential prognostic value of a decreased hemoglobin levels measured at admission due to the first episode of acute PE for post discharge all cause mortality during at least 2 years follow up.

## 2. Material and Methods

This was a prospective, single-center, follow-up, observational, cohort study of consecutive survivors of the first PE episode, managed in a single reference centre as part of the “PE-aWARE” registry (NCT03916302). The PE-aWARE (Pulmonary Embolism WArsaw REgistry) is an on-going single-centre prospective observational study of consecutive patients with confirmed acute pulmonary embolism. Its main objective is to collect and provide information on patients’ characteristics, management and outcome including short and long term survival, the frequency of chronic thromboembolic pulmonary hypertension and recurrences. The diagnosis of PE was objectively confirmed using contrast-enhanced computed tomography pulmonary angiogram when thromboembolic were visualized in at least segmental or more proximal pulmonary artery. At the admission due to acute PE transthoracic echocardiography routine laboratory tests including high-sensitivity troponin and NT-proBNP were performed. Echocardiographic examination was performed with a Philips iE 33 system (Philips Medical System, Andover, MA, USA) with 2.5–3.5 MHz transducers, within the first 24 h after admission. The dimensions of the right and left ventricles were measured in the four chamber RV focused view and R view at the level of the mitral and tricuspid valve tips in late diastole, as defined by the R wave of the continuous ECG tracing. Moreover, during hospitalization of PE index episode information was collected on PE severity according to ESC risk stratification model which included hemodynamic stability, right ventricular dysfunction detected at echocardiography or CTPA, and signs of myocardial injury assessed with elevated troponin plasma levels, and comorbidities [3] (congestive heart failure, coronary artery disease, chronic lung diseases, neoplasms). The simplified Pulmonary Embolism Severity Index (sPESI) with 1 point for each of the following: age > 80 years, history of cancer, chronic cardiopulmonary disease, pulse ≥ 110 beats/min, systolic blood pressure < 100 mmHg, oxygen saturation < 90% was calculated for every patient [10]. Patients were managed according to ESC’s current guidelines [3]. After the discharge, all PE survivors excluding moribund patients were followed for at least 24 months in our outpatient clinic. Bed ridden patients with advanced, end stage generalized cancer, requiring nursing care were regarded as moribund patients. After the acute PE phase they were transferred from our department to nursing facilities.

All patients were anticoagulated for at least 6 months. The decision to extend anticoagulation was based on the current ESC recommendations [1,3]. Briefly, anticoagulation was terminated only in patients with transient major VTE risk factor, while in subjects with unprovoked PE or when major or intermediate risk factors persisted such as active cancer, anticoagulation by default was continued undetermined unless high bleeding risk was present. During the index hospitalization or within 30 days after the discharge all patients were subjected to a routine age and gender specific screening for neoplasms. Control visits were performed in a standardized way by one of the coauthors (ODD, SP, AWC). During the first visit not only clinical status was assessed but it was also focused on results of age and sex specific cancer screening. During every control visit taking place every 6–9 months routine diagnostic laboratory tests were performed. After 6 month of anticoagulation patients reporting persistent or new onset functional limitations were referred for echocardiography. Subsequent detailed diagnostic workup was planned by managing physician.

At the end of the follow-up patients underwent a control visit or at least were interviewed by phone. Ninety patients who discontinued outpatient care in our outpatient clinic mostly due to distant residence or limitation in mobility could not be reached by phone. In this group information on clinical status, the cause and date of potential death was obtained from record of National Health Insurance which collects health records of all citizens of Poland. We recorded all-cause mortality, objectively confirmed VTE recurrences, CTEPH diagnosed according to ESC criteria [3], severe bleedings according to ISTH criteria [11], and neoplasm diagnosed during the follow-up.

## 3. Statistics

Baseline characteristics of patients are presented as parameters or median followed by interquartile range. The Shapiro–Wilk test was used to identify continuous variables with a skewed distribution which were then compared using the Mann–Whitney U test or Chi-square test. All tests were two-sided. For all performed tests *p*-values of <0.05 were considered significant. Receiver operating characteristic (ROC) analysis was used to determine the area under the curve (AUC) and the corresponding 95% confidence intervals (CIs). The prognostic relevance of analyzed parameters was assessed using univariable analysis, subsequently multivariable analysis was performed with use of logistic regression, including factors statistically important in univariate analysis Odds ratio (OR) was calculated for cutoff values identified with Youden index in ROC analysis. All analyses were performed using the STATISTICA13 data analysis software system (Dell Software, USA) or the MedCalc data analysis software system (MedCalc Software, Ostend, Belgium).

## 4. Results

### Patients Characteristics and Management

In the current study, we included 402 PE survivors after the first PE episode diagnosed and managed in our department (216 women and 186 men (aged 62.6 ± 19.51 years) and subsequently followed in our outpatient clinic for at least 2 years (Table 1). Additional 19 patients with acute PE managed in our department died during the hospital stay or were moribund patients who after the acute PE phase were transferred from our department to nursing homes and died shortly after. All those 19 patients were not included into the current study. At PE diagnosis 53% of all analyzed 402 patients had significant VTE risk factor: preexisting active neoplasm or diagnosed during the index hospitalization (18.2% patients), major trauma or surgery (34% patients). At admission intermediate risk PE was diagnosed in 74.4% of studied patients, while low risk PE and high risk in 21.4% and 4.2%, respectively. After the discharge, all patients were managed in our outpatient clinic with regular control visits. During the follow-up, new neoplasms were diagnosed in additional 13 (3.2%) patients.

All patients with detected decreased hemoglobin level below 10 g/dL on admission, underwent diagnostic work up for its causes during the index hospitalization or within 30 days after the discharge. Eventually, in 12 patients active chronic or acute bleeding was detected (6 gastro intestinal, 3 urinary tract, 3 central nervous system), while in 21 others anemia was related to chronic diseases such as chronic kidney disease or hematological disorders. During the index hospitalization, blood was transfused in order to discharge them with hemoglobin level above 9 g/dL.

After the discharge, 10% of all patients with transient major risk factors for VTE were anticoagulated for 6 months only. Moreover, in additional 3.5% subjects anticoagulation was stopped due to significant bleedings that occurred during follow-up or high bleeding risk. Additionally, 6 patients (1.5%) decided to stop anticoagulation despite physician advice. The remaining patients were anticoagulated in the long term.

In 90 patients who could not be reached by phone, clinical status was assessed with data from records of national health insurance. Due to clinically assessed very compromised prognosis based on clinical data and lack of follow-up data including type of anticoagulation they were not included in the analysis.

During a 2-year follow-up from the group of 402 consecutive PE survivors, 29 (7.2%) patients died. There were an additional 19 in-hospital deaths (all-cause 2 years mortality in 432 “all comers” was 11.4%). The latter group was not included in the analysis. Causes of 29 deaths included: severe infection or sepsis (13 cases), fatal bleeding (4 cases), progression of advanced neoplastic disease (3 cases), congestive heart failure (8 cases), and 1 case of recurrent VTE. The median time from discharge to death was 239 days varying from 14 to 1901 days. Subjects who died were older than survivors and had higher sPESI driven especially by a higher frequency of active neoplasms (37.9% vs. 16.6%, *p* < 0.001). Interestingly, elevated NT-proBNP level at admission, decreased eGFR and lower hemoglobin level characterized patients who died during the follow-up. Notably, PE severity in the acute phase assessed by ESC risk stratification model did not influence survival after the discharge. Multivariable analysis showed that only older age (*p* < 0.01) and lower hemoglobin level at admission (*p* < 0.01) were relevant for survival during the follow-up, while neoplasms were not.

We performed receiver operating characteristics analysis for hemoglobin concertation and for age in the prediction of all-cause mortality after the discharge. For hemoglobin AUC was 0.688 (95% CI 0.782–0.594), *p*-value < 0.001; Youden index based on ROC curve analysis was 13.2 g/dL. AUC for age was 0.735 (95% CI 0.651–0.819), *p*-value < 0.001 and age of 64 years was identified with Youden Index (Figure 1).

Using ROC curve we selected 2 cut-off point values of hemoglobin level. A group of 59 subjects with hemoglobin at admission below 10.5 g/dL included 10 post-discharge deaths (mortality rate 16.9%). Thus, this hemoglobin level predicted post-hospital 2-year mortality with the sensitivity 34.5% (95% CI 17.9% to 54.3%) and specificity 86.7% (95% CI 83.01–90.12%), PPV of 16.95% (95% CI 9.82–27.66%) and NPV of 5.62% (3.91–8.03%). Moreover, hemoglobin below 10.5 g/dL increased post-discharge mortality risk with OR of 4.19 (95% CI 1.82–9.65), *p*-value < 0.001. Another cut of value, hemoglobin level above 14.3 g/dL indicated low post-discharge mortality. Among 79 patients with hemoglobin level > 14.3 g/dL only one death was detected (mortality rate 1.26%). Thus, this cut off value showed high NPV of 98.73% (95% CI 91.84–99.82%), and low PPV of 8.81% (95% CI 8.13–9.53%).

Using age and hemoglobin cut off values defined by Youden index in the ROC analyses we assessed OR of all cause mortality in 4 groups. The group of patients in age ≥ 64 years hemoglobin levels < 13.2 g/dL compared to patients in the same age but with >13.2 g/dL showed OR 3.6 with 95% CI 1.3–10.1 *p* = 0.012 for death after the discharge. Moreover, when patients with age < 64 years and Hb ≥ 13.2 g/dL s were used as reference only patients in age ≥ 64 years hemoglobin levels < 13.2 g/dL showed significant OR for increased mortality (Table 2).

## 5. Discussion

The major findings of our study can be summarized as follows. During at least 2 years of follow-up of a group of 402 consecutive PE survivors 29 (7.2%) patients died. There were 19 additional in hospital deaths. All-caused 2 years mortality in 432 “all comers” was 11.4%. It should be underlined that this group of patients who died during hospital stay or were moribund and were transferred to nursing facilities was not included in further analysis. In the group discharged home hemoglobin level below 10.5 g/dL assessed at the admission identifies subjects at risk of increased mortality in long term after the discharge due to the first PE episode with OR of 4.19 (95% CI 1.82–9.65), *p*-value < 0.001. Whereas hemoglobin levels above 14.3 g/dL indicate a benign clinical course during follow-up. Moreover, since increased age and decreased hemoglobin levels were found to be significant in multivariable analysis we especially patients in advanced age with low haemoglobin level are at risk of post discharge mortality. The group of patients in age ≥ 64 years hemoglobin levels < 13.2 g/dL compared to patients in the same age but with >13.2 g/dL showed OR 3.6 with 95% CI 1.3–10.1 *p* = 0.012 for death after the discharge. We suggest that specially elderly PE survivors with decreased hemoglobin levels should be carefully supervised and followed. Austin Chin Chwan Ng et al. showed that lower serum hemoglobin and elevated troponin-T ≥ 0.1 μg/L at the time of PE are independent predictors of long-term mortality post PE [12].

It was reported that anemia, with hemoglobin levels < 13.0 g/dL in male adults and <12.0 g/dL in female adults, is an independent predictor of reduced exercise capacity, quality of life, and recurrent hospitalizations [13]. Moreover, anemia has been shown to be associated with increased mortality in both acute and chronic heart failure [14,15,16]. McCullough et al. reported that anemia in patients with heart failure is independently associated with an excess hazard for all-cause mortality and all-cause hospitalization [17]. Chronic and acute anemia lead also to poor outcomes in myocardial infarction and is a marker of an increased risk in one-year cardiovascular mortality in patients with ST elevation myocardial infarction [18,19]. In our study, patients with hemoglobin levels at hospital admission below 10.5 g/dL had the most severe prognosis, and among those whose concentration was above 14.3 g/dL, the prognosis was definitely better. There was only one death among 79 patients with hemoglobin levels above 14.3 g/dL.

NT-proBNP has been evaluated to stratify risk in patients with acute PE and remains the only biomarker that seems to be a strong predictor of 30-day prognosis [20]. Effects on short-term survival of troponin and creatinine concentrations have also been reported [21]. However, NT-proBNP was reported to be a good risk stratification marker in identifying low-risk patients who could be treated in an outpatient setting [22]. Despite the fact that NT-proBNP levels are predictors for adverse long-term outcomes in patients with known heart failure or pulmonary arterial hypertension [23], Bassan et al. showed that BNP measured at hospital admission in patients with non ST elevation acute coronary syndrome is a strong, independent predictor of very long-term all-cause mortality [24]. There is a lack of data on the effect of NT-proBNP on admission on distant complications after pulmonary embolism.

There are few reports of a post-pulmonary embolism follow-up [25,26], and there are hardly any that link data from the acute period of the disease to distant sequelae. Prognostic factors such as biomarker levels contribute to long-term morbidity after pulmonary embolism and are not fully elucidated. Although several studies have reported that among the survivors of an acute PE there is an ongoing increased risk of death long-term [27,28], current guidelines from ESC provide the same recommendation for follow-up after PE regardless of the expected survival and long-term outcome [3]. Identifying acute PE predictors of long-term mortality would allow to develop a detailed plan of care for PE patients with the worst prognosis.

In our study, nearly 400 PE patients were followed for at least 24 months. Elevated NT-proBNP and lower hemoglobin levels in the acute phase showed a significant impact on the prognosis and clinical outcomes in PE survivors.

## 6. Limitations of the Current Study

This is a single center study performed in a referral center focused on the management of venous thromboembolism both in the acute phase and in outpatient care. Causes of deaths were extracted from data of health insurance and were not adjudicated. Therefore, the results of the current study should be interpreted with caution.

## 7. Conclusions

Lower haemoglobin measured in the acute phase especially in patients in age above 64 years showed significant impact on the prognosis and clinical outcomes in PE survivors.

## Figures and Tables

**Figure 1 jcm-11-07100-f001:**
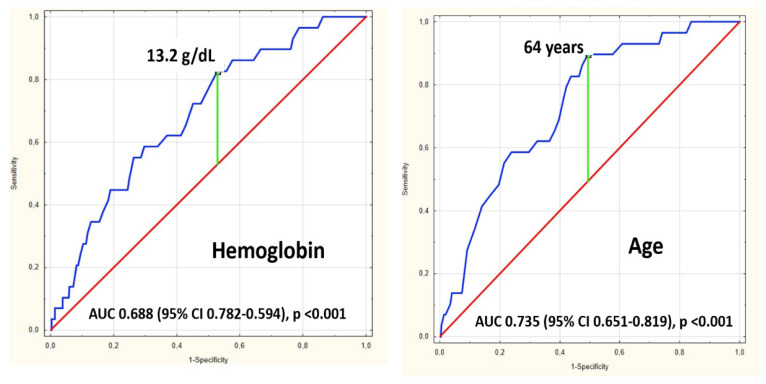
Receiver Operator Characteristics of haemoglobin concentration at admission and age of 2 year mortality in 402 PE survivors. AUC Area under the curve, CI—confidence interval.

**Table 1 jcm-11-07100-t001:** Clinical characteristics and all-cause mortality during follow-up of 402 consecutive PE survivors.

		All Patients n = 402	Survivors n = 373	Nonsurvivors n = 29	*p*-Value
Female/Male, n		216/186	197/176	19/10	0.19
Age, years		57.5 (18–97)	63 (18–97)	81 (40–93)	<0.001
Chronic heart failure, n (%)		71 (17.7%)	63 (16.9%)	8 (27.5%)	0.15
Coronary artery disease, n (%)		23 (5.7%)	20 (5.4%)	3 (10.4%)	0.25
Chronic lung disease, n (%)		45 (11.2%)	40 (10.7%)	5 (17.3%)	0.28
Active neoplasm at PE diagnosis, n (%)		73 (18.2%)	62 (16.6%)	11 (37.9%)	0.004
Neoplasm diagnosed during follow-up, n (%)		13 (3.2%)	9 (2.4%)	4 (13.8%)	<0.001
Unprovoked PE, n (%)		190 (47%)	180 (48%)	10 (34%)	0.21
Major surgery, n (%)		35 (8.7%)	33 (8.8%)	2 (6.9%)	Ns
sPESI, points		1.5 (0–5)	1 (0–5)	2 (0–4)	<0.001
	Low	86 (21.4%)	84 (22.5%)	2 (6.9%)	0.13
PE severity, n (%)	Intermediate	299 (74.4%)	273 (73.2%)	26 (89.7%)	
	High	17 (4.2%)	16 (4.3%)	1 (3.4%)	
right to left ventricular ratio > 1 in echo 4 chamber view, n (%)		101 (32.4%)	94 (32.7%)	7 (29.2%)	0.72
LV EF (%)		60 (15–70)	60 (15–70)	60 (20–65)	0.08
Troponin (μg/L)		0.0175 (0.003–1.59)	0.038 (0.003–1.59)	0.074 (0.01–0.8)	0.17
NT-proBNP (pg/mL)		3710.5 (2–28,879)	344.5 (5–28,879)	1440 (74–12,330)	0.004
D-dimer (µg/L)		19,050 (2–111,459)	4558 (2–111,459)	4613 (580–26,945)	0.79
Hemoglobin at admission (g/dL)		10 (3.1–19.3)	13.1 (3.1–19.3)	11.7 (6–14.8)	<0.001
Plasma creatinine (mg/dL)		1.04 (0.33–6.5)	0.9 (0.33–5.4)	1 (0.48–6.5)	0.36
estimated glomerular filtration rate (CockroftGault, mL/min)		89.18 (9.2 ≥ 100)	80.11 (10.9 ≥ 100)	59.52 (9.2 ≥ 100)	0.03

PE—pulmonary embolism, sPESI simplified Pulmonary Embolism Severity Index.

**Table 2 jcm-11-07100-t002:** Prognostic value of age and hemoglobin levels in PE survivors.

	n	Deaths	Mortality	OR	OR When Group with Age < 64 Years and Hb > 13.2 g/dL as Reference
Age < 64 years and Hb > 13.2 g/dL	97	1	1.03%	2.1 95% CI 0.2–23.7, *p* = 0.54	1 as reference
Age < 64 years and Hb < 13.2 g/dL	93	2	2.15%		2.1 95% CI 0.2–23.7, *p* = 0.54
Age > 64 years and Hb > 13.2 g/dL	89	5	5.62%	3.6 95% CI 1.3–10.1 *p* = 0.012	5.7 95% CI 0.7–49.9, *p* = 0.11
Age > 64 years and Hb < 13.2 g/dL	118	21	17.80%		20.8 95% CI 2.7–157.6, *p* = 0.003

## Data Availability

Not applicable.

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
