# Peer review of "Decreased Haemoglobin Level Measured at Admission Predicts Long Term Mortality after the First Episode of Acute Pulmonary Embolism"

_jcm, 2022, doi:10.3390/jcm11237100_

Round 1

Reviewer 1 Report

Authors of the manuscript   Hemoglobin measured at admission predicts long term mortal-2 ity after the first episode of acute pulmonary embolism. The topic is interesting, but the article needs improvement.

Comments:

Title of manuscript: The title of the manuscript should show the achievement of the aim of the work - I suggest that the title should contain an explanation of what was studied in the paper

Goal of the study: There is no clearly defined purpose of the paper- it should be placed at the end of the introduction

Methods section:

Additional clarification is needed regarding the "PE-aWARE" registry (NCT03916302)-inclusion/exclusion criteria, project of the registry- study parameters

Clinical parameters:

Echocardiography: Were only the sizes of the left and right ventricles analyzed? Even the LVEF rating is missing.

Laboratory parameters: Why in the analysis of the morphology parameters only the hemoglobin level was taken into account? It would also be interesting to compare the parameters of inflammation and platelets.

The methodology also lacks a detailed description of the follow-up visits - how many people during the 2-year follow-up period underwent control laboratory tests and echocardiography? For how many people was it just a follow-up by phone? How many patients were excluded from the study due to the lack of follow-up

Results

The study does not present detailed and clinical characteristics of the patients. The number of clinical parameters studied is very small, which makes it practically impossible to draw conclusions about predictive factors of long-term mortality. How many patients have comorbidities - e.g. diabetes, severe heart failure with low LVEF, previous surgery?

In the description of statistical methos we find the sentence: “The prognostic relevance of analyzed parameters was assessed using univariable  analysis, subsequently multivariable analysis was performed for all parameters found significant in the univariable model”- meanwhile, the results section did not present the results of the multivariate analysis

The manuscript lacks a graphic representation of the results of ROC curve

Author Response

Reviewer 1.

  1. Title of manuscript: The title of the manuscript should show the achievement of the aim of the work - I suggest that the title should contain an explanation of what was studied in the paper. Reply: Thank you for this comment. Following this recommendation we have modified the title of the manuscript as follows Decreased hemoglobin level  measured at admission predicts long term mortality after the first episode of acute pulmonary embolism”.
  2. Goal of the study: There is no clearly defined purpose of the paper- it should be placed at the end of the introduction Reply: following this comment we  clarified the purpose of our paper in the end of the section Introduction   line 41 and 80 “ We hypothesized that hemoglobin levels may be   useful for post discharge prognostication  after the first episode  of acute pulmonary embolism.  Therefore,  the aim of the current study was to  evaluate a  potential prognostic value of  a  decreased   hemoglobin levels measured at admission  due  to  the first  episode of acute PE for post discharge all cause mortality during at least  2 years follow up.. We have also  modified  accordingly the abstract .
  3. Additional clarification is needed regarding the "PE-aWARE" registry (NCT03916302)-inclusion/exclusion criteria, project of the registry- study parameters Reply: Following this comment we provided more details on PE-aWARE" registry . Line 89: The PE-aWARE (Pulmonary Embolism WArsaw REgistry) is an on-going single-centre prospective observational study of  consecutive  patients with confirmed acute pulmonary embolism (APE). Its main objective is to collect and provide information on patients' characteristics, management and outcome  including  short and long term  survival, the frequency of thromboembolic pulmonary hypertension  and recurrences.
  4. Echocardiography: Were only the sizes of the left and right ventricles analyzed? Even the LVEF rating is missing. Reply: All studied  patients  at the admission   underwent echocardiographic examination which included not only  assessment of the right heart, but the left heart with LVEF. Following this comment  LVEF is now  presented in the table    Please  note that  there were no differences  in LVEF between   survivors and non – survivors  60% (15-70) vs 60% (20-65), p 0,08.
  5. Laboratory parameters: Why in the analysis of the morphology parameters only the hemoglobin level was taken into account? It would also be interesting to compare the parameters of inflammation and platelets. Reply: Indeed,  we fully  agree that it would be interesting to analyze additional  laboratory parameters  including platelets and inflammatory markers such as   However, in the current study we  have focused on  hemoglobin,  D-Dimer levels, troponin, and  NTproBNP.  We feel  that   our findings  on prognostic value of hemoglobin  levels are still  of potential value  and  interesting   and  would suggest not to  extend them. 
  6. The methodology also lacks a detailed description of the follow-up visits - how many people during the 2-year follow-up period underwent control laboratory tests and echocardiography? For how many people was it just a follow-up by phone? How many patients were excluded from the study due to the lack of follow-up. Reply:  Thank you for this comment.  All patients with acute PE managed in your department are routinely  followed in our outpatient clinic  with the exception of  bed ridden patients with severely limiting comorbidities such as advanced cancer.  Control visits were performed in a standardized way  by one of the coauthors  (ODD, SP, AWC).  During the first visit  not only  clinical status is analyzed, but   it is focused on results of  age and sex specific cancer     During all control visits every 6-9 months routine diagnostic laboratory tests were performed. After 6 month of anticoagulation patients reporting persistent or new onset  functional limitations were referred   for echocardiography. Subsequent detailed diagnostic workup was planned  and  coordinated  by  the managing physician.  At the end of  at least 2  years follow up patients underwent a control visit or were interviewed by phone.  Indeed, 90 patients  discontinued   outpatient care in our outpatient clinic mostly due to  distant residence or limitation in mobility. In this group information on clinical status,  the cause and date of  potential  death was obtained from record of National  Health Insurance  which  collects  detailed  health  records of all citizens of  Poland.    Following this comment paragraph on   follow up was modified: line 124:   “Control visits were performed in a standardized way  by one of the co-authors  (ODD, SP, AWC).  During the first visit  not only   clinical status was assessed but   it  was  focused on results of  age and sex specific cancer  screening.   During  every  control visit taking place every 6-9 months  routine diagnostic laboratory tests were performed.   After 6 month of anticoagulation patients reporting persistent or new onset  functional limitations were referred  for echocardiography. Subsequent detailed diagnostic workup  was planned by managing physician . At the end of the follow-up patients underwent a control visit or at least were interviewed by phone. Ninety patients   who  discontinued  outpatient  care in our outpatient clinic mostly due to  distant residence or limitation in mobility could not be reached by phone. In this group information on clinical status,  the cause and date of  potential  death was obtained from record of National  Health Insurance  which  collects   health  records of all citizens of  Poland.”  Moreover  we have clarified the paragraph on moribund patients  line 158 Additional 19 patients with acute PE managed in our department died during the hospital stay  or  were  moribund patients  who after the acute PE  phase   were transferred  from our department  to nursing homes and died  shortly after. All these 19  patients were not included into the current study”.   
  7. The study does not present detailed and clinical characteristics of the patients. The number of clinical parameters studied is very small, which makes it practically impossible to draw conclusions about predictive factors of long-term mortality. How many patients have comorbidities - e.g. diabetes, severe heart failure with low LVEF, previous surgery? Reply: Thank you for this  Clinical data of the studied patients are presented in the table 1 and include heart failure, coronary artery disease, chronic lung disease.    We also present  results of sPESI usually used for prognostication in PE patients. Following this comment we have added LVEF   which was statistically  not different between survivors and non – survivors  60% (15-70) vs 60% (20-65), p 0,08.   In addition,  there were no  differences in  frequency of patients with  severely  reduced LVEF (<30%)   between  aforementioned groups 3 vs 1  case .  Moreover, we have  analyzed  major surgery performed  within 30 days before PE episode.    8,8% of survivor and 6,9 % no survivors  underwent surgery (ns). Table 1.   
  8. In the description of statistical methods we find the sentence: “The prognostic relevance of analyzed parameters was assessed using univariable  analysis, subsequently multivariable analysis was performed for all parameters found significant in the univariable model”- meanwhile, the results section did not present the results of the multivariate analysis. Reply: Please note that  in line  195 we indicatedMultivariable analysis showed that only older age (p<0,01) and lower hemoglobin level at admission (p<0,01) were relevant for survival during the follow-up, while neoplasms were not”.
  9. The manuscript lacks a graphic representation of the results of ROC curve Reply: Following this recommendation we have added figure  with  graphic representation of the results of ROC curve for age an hemoglobin levels for  2 years survival.

Reviewer 2 Report

Line 33: What does 'integrated patient care' mean? Perhaps the authors can elaborate how their results can improve patient care in detail in the discussion.

Line 55: Please be more specific about the model - include parameters and what it is stratifying for (?haemodynamic instability?)

Line 57: Include reference for the sPESI and also the criteria in an appendix.

Line 59: How was 'moribund' defined? 

Line 68: How long was the follow-up period? Were the 90 patients who could not be reached by phone included in final analysis?

Line 84: How was the final multivariate model arrived at? How did you select variables to include in the search for multivariate model? What kind of regression was used for the multivariable analysis? More detailed statistical methods required. Given there was such a high variance between discharge to death (ranging up to 1901 days!), time dependent analysis is recommended (ie COX-regression). 

Line 92: A significant number of patients died during the initial hospitalisation (19 patients). It would be of interest to the reader to describe the characteristics of these patients - were they different to other groups? what were their sPESI scores? Were these patients included in your final multivariate model? Excluding these patients from the model would be a major flaw as you are using Hb at diagnosis (and not at discharge from hospital).

Line 108: does this mean 90% of patients went on to continue indefinite anticoagulation?

Table 1: Nonsurvivors have lower eGFR than survivors? Please check this

Line 127: Did ESC or PESI stratify for patients who died in hospital? (the 19 patients).

Line 131: What was the ROC AUC of your final multivariate model? (ie. age + Hb). Another table with variate co-efficients or OR/HR and p-values is required.

Line 144: The AUC of Hb is less than that for age, not surprisingly. How much additional predictive value did Hb add to the effect of age?

Finally, anaemia may reflect anaemia of chronic disease (and therefore multiple co-morbidities), bleeding, cancer, anticoagulant use, bone marrow pathology and old age). Suggest authors to include in the discussion this interpretation of why Hb should be in their final model.

Author Response

Reviewer 2

  1. Line 33: What does 'integrated patient care' mean? Perhaps the authors can elaborate how their results can improve patient care in detail in the discussion. Reply:  Contemporary  ESC guidelines  on acute pulmonary embolism  (EHJ 2019)  underline  that  an integrated model of patient care after PE should be provided.  It should include   interdisciplinary care  and standardized treatment protocols.  All patients after acute PE managed in our department are routinely  followed in our outpatient clinic  with the exception of  bed ridden patients with severely limiting comorbidities such as advanced cancer.  Control visits are  performed in a standardized way.  During the first visit  not only  clinical status is analyzed, but   it is focused on results of  age and sex specific cancer  During all control visits every 6-9 months routine diagnostic laboratory tests were performed. After 6 month of anticoagulation patients reporting persistent or new onset  functional limitations were referred   for echocardiography. Subsequent detailed diagnostic workup  was planned by managing physician.    Following this comment we have added the sentence line 72 “Therefore, in order to optimize  patients follow-up   after acute PE it is recommended to ensure an integrated patient care after PE  which should include   interdisciplinary  standardized  management and treatment”.
  2. line 55: Please be more specific about the model - include parameters and what it is stratifying for (?haemodynamic instability?) Reply: Following this comment we defined ESC stratification model.  Therefore following statement has been added line 102  Moreover, during hospitalization of PE index episode information was collected  on PE severity according to ESC risk stratification model which included hemodynamic stability, right ventricular dysfunction  detected at echocardiography or CTPA, and  signs of myocardial injury assessed with  elevated troponin  plasma levels, and comorbidities  3.
  3. Line 57: Include reference for the sPESI and also the criteria in an appendix. Reply: following with recommendation we have added as follows line 107 “The simplified Pulmonary Embolism Severity Index (sPESI)  with  1 point for each of the following: age >80 years,  history of cancer, chronic cardiopulmonary disease, pulse >110 beats/min, systolic blood pressure <100mmHg, oxygen saturation <90%  was calculated for every patient 10.
  4. Line 59: How was 'moribund' defined? Reply: Bed ridden patients with advanced, end stage generalized cancer, requiring  nursing  care were regarded as moribund patients.  After the acute PE  phase  they were   transferred  from our department  to nursing home.    Following this comment we  provided a  description of moribund patients line 112  “Bed ridden patients with advanced, end stage generalized cancer, requiring  nursing  care were regarded as moribund patients.  After the acute PE  phase  they were   transferred  from our department  to nursing home”    
  5. Line 68: How long was the follow-up period? Were the 90 patients who could not be reached by phone included in final analysis? Reply: All patients were followed for 2 years . Indeed 90 patients  discontinued   outpatient care in our outpatient clinic mostly due to  distant residence or limitation in mobility . Following sentence was added  line 130  “Ninety patients   who  discontinued  outpatient  care in our outpatient clinic mostly due to  distant residence or limitation in mobility could not be reached by phone. In this group information on clinical status,  the cause and date of  potential  death was obtained from record of National  Health Insurance  which  collects   health  records of all citizens of  ”
  6. Line 84: How was the final multivariate model arrived at? How did you select variables to include in the search for multivariate model? What kind of regression was used for the multivariable analysis? More detailed statistical methods required. Given there was such a high variance between discharge to death (ranging up to 1901 days!), time dependent analysis is recommended (ie COX-regression). Reply:  into  multivariable model we included  parameters found to be  statistically significant in  univariable analysis (age,  cancer, sPESI,  NTproBNP , Hb at admission eGFR).  Only age and  hemoglobin levels were found to significant in  multivariable linear regression analysis.
  7. Line 92: A significant number of patients died during the initial hospitalization (19 patients). It would be of interest to the reader to describe the characteristics of these patients - were they different to other groups? what were their sPESI scores? Were these patients included in your final multivariate model? Excluding these patients from the model would be a major flaw as you are using Hb at diagnosis (and not at discharge from hospital). Reply: Thank you for this   Please find below characteristics of  19 patients who  died.  They were older,  had higher sPESI score and presented  higher  PE severity - 57,9% presented high risk  PE.  

Patients

N=19

Female/Male

11/8

Age

81,9 (59-100)

Chronic heart failure

5 (26,3%)

Coronary artery disease

4 (21%)

Chronic lung disease

4 (21%)

Active neoplasm at PE diagnosis

5 (26,3%)

sPESI

2 (0-5)

Low

0 (0%)

PE severity

Intermediate

8 (42,1%)

High

11 (57,9%)

right to left ventricular  ratio

> 1 in echo 4 chamber view

5 (26,3%)

They were not included into the analysis because the aim  of the current study was to  evaluate  prognostic role of hemoglobin in  patient who survived the acute PE episode  and were discharged home .    

  1. Line 108: does this mean 90% of patients went on to continue indefinite anticoagulation? Reply:  Indeed, after the discharge, only  10% of all patients with transient major risk factors for VTE were anticoagulated for 6 months only.  In the remaining subjects there was at least intermediate risk pf PE recurrence.  Following contemporary ESC guidelines (recommendation IIa)  they were advised   to continue    In approx. 20% of them after 6 months of initial treatment  the   intensity of anticoagulation was decreased for example apixaban  2,5mg  bid or  rivaroxaban 10md daily . In additional 3,5% subjects anticoagulation  was stopped due to significant bleedings that occurred during follow-up or high bleeding risk. Additionally, 6 patients (1,5%) decided to stop anticoagulation despite physician advice. The remaining patients were anticoagulated in the long term.
  2. Table 1: Nonsurvivors have lower eGFR than survivors? Please check this. Reply: Although plasma creatinine level was similar between survivor and non survivors  eGFR calculated with CG formula  was lower in  non-survivors  than survivors 59,52 (9,2->100) vs 80,11 (10,9->100 p=0,03. Creatinine level and eGFR  are presented in table 1.
  3. Line 127: Did ESC or PESI stratify for patients who died in hospital? (the 19 patients). Reply : Indeed 19 patients who died during  hospitalization presented  more severe PE  according to ESC stratification strategy  - 57,9% of the were diagnosed with high risk PE while  in survivors only 4,2 % were diagnosed with high risk PE (p<0.01).  Moreover, sPESI was higher in this  group  which was  mainly driven by advanced age (81,9 yr)     
  4. Line 131: What was the ROC AUC of your final multivariate model? (ie. age + Hb). Another table with variate co-efficients or OR/HR and p-values is required. Line 144: The AUC of Hb is less than that for age, not surprisingly. How much additional predictive value did Hb add to the effect of age? Reply: Thank you for  this comment. We assessed OR   between  4 groups  formed   according to age and hemoglobin levels  defined by Youden index in the ROC analysis, In the group of patients in age > 64   years  hemoglobin levels < 13,2  g/dL   showed OR    3,6    with 95%CI 1,3-10,1 p=0,012 for death after the discharge. Moreover,  when   patients   with age <64 years  and Hb>13,2g/dL were used as reference   only of patients in age > 64   years  hemoglobin levels < 13,2  g/dL were  at risk  of increased mortality.   

Table 2 was added and discussed in the text

N

Deaths

mortality

OR

OR

Age <64 years  and Hb>13,2g/dL

97

1

1,03%

Ns

1  as reference

Age <64 years and Hb<13,2g/dL

93

2

2,15%

Ns

Age >64 years and  Hb>13,2 g/dL

89

5

5,62%

3,6 

95%CI 1,3-10,1 p=0,012

Ns

Age >64 years  and  Hb<13,2g/dL

118

21

17,80%

20,8 

95% CI 2,7-157,6, p=0,003

  1. Finally, anaemia may reflect anaemia of chronic disease (and therefore multiple co-morbidities), bleeding, cancer, anticoagulant use, bone marrow pathology and old age). Suggest authors to include in the discussion this interpretation of why Hb should be in their final model. Reply: Thank you for this comment. In order to avoid the influence of anticoagulant treatment on the hemoglobin concentration, the value of hemoglobin found on admission was used for the analysis. Indeed, the concentration of hemoglobin depends on comorbidities, but in the univariate analysis only the neoplastic disease had a significant impact on survival, while in the multivariate analysis only age and hemoglobin concentration reflected on the prognosis.

Round 2

Reviewer 1 Report

The authors provided sufficient explanations

Author Response

Thank you very much for this nice comment.

Reviewer 2 Report

Thank you for the changes, the manuscript is now much easier to understand. I have just 2 minor points to clarify:

1. Table 2: why are there 2 columns with OR as heading? It is not clear the difference. Also even if ORs were not significant, the OR and p-value should be displayed in the table. 

2. Were the 90 patients lost to follow up included in the final 402 patients? This information should be moved to the results section. If these patients were included in the 402 patients, they will need to be included in Table 1.

Author Response

Thank you for your comment.

Following your recommendations, Table 2 has been modified. OR and p values and explanation for OR in the last column have been added.  

In 90 patients who could not be reached by phone, clinical status was assessed with data from records of national health insurance. Due to clinically assessed very compromised prognosis based on clinical data and lack of follow-up data including type of anticoagulation they were not included in the analysis.

In 90 patients who could not be reached by phone, clinical status was assessed with data from records of national health insurance. Due to clinically assessed very compromised prognosis based on clinical data and lack of follow-up data including type of anticoagulation they were not included in the analysis. We have added this sentence to the Results section.